# Gender Difference in the Effect of Short Sleep Time on Suicide among Korean Adolescents

**DOI:** 10.3390/ijerph16183285

**Published:** 2019-09-06

**Authors:** Woong-Sub Park, SangA Kim, Hyeyun Kim

**Affiliations:** 1Department of Preventive Medicine & Public Health, College of Medicine, Catholic Kwandong University, Gangwon 25601, Korea; 2Department of Social Welfare, Dong Seoul University, Gyenggi 13117, Korea; 3Department of Neurology, College of Medicine, Catholic Kwandong University, International St. Mary’s Hospital, Incheon 22711, Korea

**Keywords:** gender, sleep time, suicide, Korean, adolescents

## Abstract

A close association between the duration of sleep and suicide has been reported in previous studies. This study was designed to investigate whether there is a difference in the effects of sleep duration on suicide by gender. This study was conducted based on the results of a volunteer online survey for adolescents in middle and high school in the Republic of Korea. The results showed that the effect of a depressive mood on short sleep time and on suicide was not different between male and female adolescents. It has been reported that the direct effect of short sleep time on increasing suicidal ideation is 2.50 times higher in female than in male adolescents. Differences in the metabolism of sex hormones and sleep-associated neurotransmitters might have affected this result, but further studies are needed to clarify more obvious mechanisms. In addition, this result should be considered when establishing sleep education for adolescents.

## 1. Introduction

The physiologic role of sleep is not limited to physical growth but also plays an important role in mental development in adolescents [1]. An important and desirable function of sleep is to aid in the recovery of adolescents exposed to emotional stress, while sleep deprivation and abnormal sleep patterns have adverse effects on mental health [2].

Several reports have shown the differences in sleep duration between women and men [3]. The combination of environmental, social, and cultural influences on male and female biological factors contributes to gender differences in sleep [4]. Hormonal and physical changes during certain periods, such as puberty, pregnancy, and menopause, could affect sleep health during a woman’s lifetime [5]. The risk of insomnia could begin with menarche, a period in which huge hormonal changes begin [6]. Gender differences exist in sleep quality, duration, the latency of sleep onset, and sleep structure in the general population [7].

We have previously reported, for female adolescents, that sleep duration has a direct effect on suicidal ideation, regardless of other factors, such as economic status, perceived health status, depression, and so on [8]. The association between short sleep time and depression/suicidal ideation has been reported in several studies [9,10,11]. We designed this study by paying attention to whether there was a difference in the impact of short sleep time on suicide between male and female adolescents. This study extended the target group to high school students in order to examine the direct and indirect effects of short sleep time on depression and suicide according to gender and analyzed the results with more recent data. 

## 2. Methods

This study was conducted by the Korea Center for Disease Control and Prevention (KCDC). Data from the juveniles who responded to the 13th juvenile health behavior online survey on sleeping time in 2017 were analyzed. This online survey was an anonymous, voluntary online survey for adolescents in middle school and high school that aimed to understand youth health behaviors, such as smoking, alcohol consumption, obesity, diet, and physical activity in the Republic of Korea. Moreover, this government-approved (approval No. 117058) and KCDC institutional review board-approved (2014-06EXP-02-P-A) statistical survey has been conducted annually since 2005. All subjects provided consent to participate in the study. The 2017 survey included 123 items from 15 categories, including smoking, alcohol consumption, obesity, diet, and physical activity, and 107 indicators were calculated. The questionnaire items and indicators were developed with the help of expert advisory committees.

Sampling of online health behaviors was conducted using a stratified cluster analysis. First, data were extracted according to school and then according to class. During the first stage of data extraction, a list of the population was sorted, the sampling interval was calculated, and the sample school was selected using a systematic sampling method. During the second stage of data extraction, data were randomly selected from the selected sample schools (one class per grade). Of the 64,991 adolescents surveyed from 800 schools (400 middle schools and 400 high schools), 62,276 adolescents participated in the survey (95.8%). In the Republic of Korea, the education system consists of six years of elementary school, three years of middle school, and three years of high school. For descriptive convenience, the grades from the first grade of middle school to the third grade of high school are described as Grade 1 to 6. This study analyzed 49,152 adolescents in middle and high school. The definition of each variable was similar to that of the 2017 Youth Health Behavior Online Survey. The main variables were depressive experience (i.e., sadness or despair that affected daily living for two consecutive weeks during the past 12 months) and suicidal ideation (i.e., seriously considering suicide during the past 12 months). The average sleep duration per night was categorized as <5 h, >5 h but <6.5 h, >6.5 h but <8 h, and >8 h, and this was analyzed as an independent variable. The variables that could affect the mental health of female adolescents were economic status (upper class, middle class, or lower class); living with one parent, both parents, or neither parent; current smoking status (i.e., smoking more than one cigarette per day during the past 30 days); current alcohol consumption (i.e., consuming more than one drink during the past 30 days); and perceived health status. Smartphone use per day during weekdays, except for academic purposes, was categorized as <1 h, >1 h but <2 h, >2 h but <4 h, and >4 h. Sleep duration, smartphone use, and caffeine consumption were analyzed as continuous variables in path analysis.

For the online survey, a composite sample with a weighted value was designed to represent the youth in Korea. The KCDC provided guidelines for the raw data to help analyze the weighted value and appropriately reflect the sample design information. Therefore, all analyses used a method provided by the KCDC that considered the weight of the composite sample.

We used the SAS version 9.3 (SAS Institute, Cary, NC, USA) statistical program to perform a bivariate analysis of the general characteristics, depressive experiences, and suicidal ideation. Then, we analyzed the mediating effects of depressive experiences on the influence of sleep duration on suicidal ideation using the M-plus version 5.2 (Muthén & Muthén, Los Angeles, CA, USA) statistical package. 

## 3. Results

### 3.1. General Analysis

Finally, 49,152 middle and high school adolescents (female 24,781 and male 24,371) participated in the study. Table 1 shows the general characteristics of all participants. All participants were uniformly distributed in each school grade. The respondents who answered that their economic level was low were 12.89% for female and 12.25% for male adolescents. The percentage of those not living with their parents was 0.99% for female and 0.89% for male adolescents. The smoking rate was 2.26% for female, and 7.79% for male, adolescents—higher than for female adolescents. In the alcohol consumption survey, the female adolescents were 12.16% and the male adolescents were 16.15%; the alcohol consumption ratio was higher than the smoking rate in these adolescents. In a questionnaire on subjective health condition, 32.65% of females and 22.71% of males said that they thought they were not healthy. A total of 49.25% of female adolescents and 35.66% of male adolescents answered that they use smartphones for more than four hours on a weekday for non-academic purposes. Drinking caffeine-containing drinks at least five times a week in the questionnaire was 2.6% for female and 3.18% for male adolescents.

In the question about sleep duration, 21.03% of female adolescents and 12.26% of male adolescents answered that they sleep for less than five hours a night. The proportion of female adolescents who answered that they would sleep for more than eight hours was 10.77%, and this was 20.70% for male adolescents (Table 2). The proportions of depression were 29.72% for female adolescents and 19.84% for male adolescents. In the question on suicidal ideation, 14.8% of female adolescents and 9.03% of male adolescents answered that they had thought about suicide (Table 2).

The risk factors for depression (Table 3) and suicidal ideation (Table 4) were analyzed. In the results of the analysis of risk factors for depression, the higher the grade, the worse the depression in both genders. The lower the economic level, the more severe the depression was for both male and female adolescents. In the smoking, alcohol consumption, and caffeine consumption groups, depression was higher than in the other groups for both genders. The shorter the sleep duration, the higher the degree of depression. 

In the results, adolescents of both genders had the most frequent depression in 2nd graders of high school, while the 2nd graders of middle school had the highest prevalence of suicidal ideation. Low economic level was a factor that increased the frequency of depression and suicidal ideation in both genders. Living with one’s parents did not affect instances of depression and suicidal ideation. Alcohol consumption, smoking, perceived poor health, using smartphones for 4 h or more per day, and drinking caffeine-containing drinks at least five times a week were factors that affected depression and suicidal ideation for both genders. In sleep duration analysis, a sleeping time of less than 5 h in males and females was confirmed to be a major factor in both depression and suicidal ideation.

### 3.2. Path Analysis Results

In order to confirm the association of suicidal ideation and depression with the correlated risk factors between female and male adolescents, path analysis was used to adjust the other factors affecting depression and suicide. Table 5 presents the results of analyzing the direct effects of several variables on depression and suicidal ideation by gender. The standardized direct effects of sleep duration on suicidal ideation in female and male adolescents were −0.039 and −0.021, respectively. In addition, the standardized direct factor analysis of smartphone use on depression showed a 2.3 times higher result for male than for female adolescents (0.075 for females and 0.033 for males). Caffeine consumption also showed a gender difference. A standardized direct factor analysis gave a value of 0.016 for female adolescents and a value of 0.038 for male adolescents. Thus, caffeine consumption had a higher impact on males than females (Table 5). We also analyzed the indirect and total effects of reduced sleep duration on suicidal ideation, as well as the indirect effects via depression. 

The direct effect of short sleep on suicidal ideation was 1.24 times higher in female than in male adolescents. However, the indirect effects were 1.02 times higher for female than male adolescents (Table 6). The standard coefficients and unstandardized coefficients of the variables for both genders are summarized in Figure 1. Figure 1 shows the effect of reduced sleep time on suicidal ideation via a depressive mood—that is, the indirect effects did not show differences between male and female adolescents, but the direct effect of reduced sleep time on suicide was bigger for females than for males.

## 4. Discussion

The most significant result in our study is that the direct effect of short sleep on suicide was 2.5 times higher for females than for males. Previous studies have also reported that a lack of sleep duration increases suicide among adolescents [12,13]. While there are differences depending on the different culture in which the study was conducted, a sleep time of less than seven or eight hours commonly increases the rate of suicide [12,13]. However, no analysis of gender differences was made in these previous studies. The direct effect of sleep time on depression and the indirect effect of sleep time on suicide via depression did not show meaningful differences between males and females. Only the direct effects of the standardized coefficients were obviously different between males and females, while the indirect effects on suicide showed no significant difference between the two genders.

In this study, the impact of sleep time on suicide for male and female adolescent was shown. The results suggest several hypotheses. First, there were gender-dependent differences between males and females in their hypothalamic-pituitary-adrenal (HPA) axis stress responses [14]. In stress situations, males had greater adrenocorticotropic hormone (ACTH) and cortisol levels than females [15]. Gender-related differences in HPA axis stress reactivity could be related to the differences in the prevalence of mood disorders, such as anxiety and major depressive disorder [16]. For males, defenses against emotional stress conditions through the HPA axis response and hormone secretion may be stronger than in females. The development of defense mechanism related hormones in males may have helped males adapt to various emotional stress situations, including sleep deprivation [17]. Secondly, leptin in the hypothalamus is a key neurotransmitter related to sleep physiology. Recent studies have shown that dysregulation of leptin may be related to mechanisms of psychopathology, including various emotional changes, such as anxiety, depression, sleep disorders, and suicide [18,19]. In addition, the leptin receptors and their projections in the amygdala, which is a major structure for anxiety and stress-related emotional changes, offer another basis for the association with emotional changes [20]. Females, compared to males, show increased levels of morning plasma leptin after sleep restriction [21]. This could explain the results of our study, which showed that the effect of sleep duration on suicide is greater in female adolescents. That is, in the case of sleep deprivation in women, the concentration of the plasma level of leptin increases more, and this increase stimulates the amygdala to control changes in anxiety and emotional stress. Leptin and leptin-related neuro-circuits are activated in stress situations, and a decrease in the leptin level results in a decrease of the HPA axis-related hormone, which could play a protective role during stressful conditions [22]. Based on this scientific mechanism, decreased leptin levels in female adolescents could be a sleep deprivation risk factor for those vulnerable to suicide. Third, the level of serum melatonin showed gender differences independent of light exposure [23,24]. There is no study of the differences in gender-specific melatonin levels among adolescents, but it has been reported that female adults have lower levels than males [23]. Melatonin is a major sleep-related hormone and has been studied in relation to depression and suicide [25,26,27]. This is the reason why melatonin supplementations are not only used for sleep disorders, such as insomnia, but also as an adjunctive therapy for psychiatric disorders [28]. These studies have shown that melatonin plays a positive role in maintaining emotional stability, as well as controlling sleep and circadian rhythms. The protective effects of melatonin on emotion suggest that decreased levels of melatonin in women could be one of the causative mechanisms behind our results, which show gender differences for the effects of sleep deprivation on suicide.

In addition to duration of sleep, there are other factors affecting depression and suicide that showed a gender difference. The effect of alcohol consumption on suicidal ideation was 2.4 times higher among female than male adolescents. Previous studies have reported the relationship between alcohol consumption and depression [29]. In a single study conducted in the United States, alcohol consumption was an independent cause of depression in male adolescents but reported to have bidirectional effects in female adolescents [30]. It is not clear whether alcohol intake is a cause or a consequence of depression, but previous studies also reported a close association in females. This is similar to the results of this study, which indicates that there is a stronger direct effect on suicide based on alcohol consumption in females than in males.

In summary, the effects of physical health, the excessive use of smartphones, alcohol drinking, and short sleep duration on mental health, including a depressive mood and suicidal ideation, are more powerful for females than males.

This study has some limitations. First, during the design step for sampling, to involve large-scale adolescent groups, adolescents of the correct age who did not attend school were excluded because sampling was based within schools. The main limitation of this study is that the adolescents who do not attend school for various reasons may be a more economically, physically, and mentally vulnerable group [31]. Second, we analyzed only sleep time; our study on the qualitative aspects of sleep was insufficient. In order to confirm the mechanism of the association between sleep and depression/suicide, consideration of the quality of sleep is needed. Third, this is a cross-sectional questionnaire-based design, for which there could be fatal limitations in the relationship between cause and outcome. We used the path analysis method to compensate for this limitation. Follow-up studies suggest that this limitation could be overcome. 

Despite its limitations, this is the first study to analyze the relationship between gender and the effects of reduced sleep time on suicidal ideation in adolescents. The major results from this study might be considered when planning sleep education for adolescents. Especially in female adolescents, short sleep time was affected by negative emotions, such as depression and suicidal ideation. When deciding on policies for education in middle and high school, it may be necessary to take gender differences into consideration.

## 5. Conclusions

In conclusion, we suggest that the direct effect of short sleep time on suicidal ideation in female is stronger than male adolescents. The result of this study should be considered in future education for sleep and mental health in adolescents. 

## Figures and Tables

**Figure 1 ijerph-16-03285-f001:**
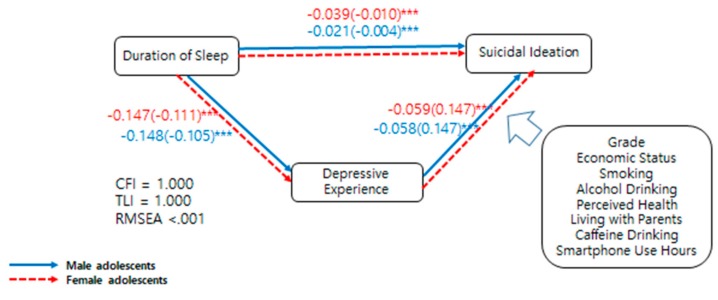
Standardized coefficients (unstandardized coefficients) of variables from the estimated structural equation model in male adolescents (blue line) and female adolescents (red line). CFI: comparative fit index; TLI: Tacker–Lewis index; RMSEA: root mean square error of approximation; *** appears statistically significant (*p* < 0.01).

**Table 1 ijerph-16-03285-t001:** General characteristics.

Variables		Female			Male		
		*N*	Weighted (%)	S.E. of %	*N*	Weighted (%)	S.E. of %
**Grade**	Middle school 1	4362	15.89	0.52	4315	15.92	0.51
	Middle school 2	4481	16.46	0.55	4379	16.53	0.54
	Middle school 3	4576	16.32	0.55	4389	16.55	0.56
	High school 1	3767	16.05	0.58	3510	15.64	0.60
	High school 2	3842	17.71	0.64	3959	17.87	0.62
	High school 3	3753	17.57	0.67	3819	17.49	0.60
Economic status	Middle High	21,530	87.11	0.34	21,324	87.75	0.31
	Low	3251	12.89	0.34	3047	12.25	0.31
Living with parents	No	255	0.99	0.08	239	0.89	0.08
	Yes	24,267	99.01	0.08	23,781	99.11	0.08
Current smoking status	No	24,233	97.74	0.13	22,571	92.21	0.27
	Yes	548	2.26	0.13	1800	7.79	0.27
Current alcohol consumption status	No	21,929	87.84	0.39	20,569	83.85	0.35
	Yes	2852	12.16	0.39	3802	16.15	0.35
Perceived health status	Poor	7957	32.65	0.40	5453	22.71	0.34
	Good	16,824	67.35	0.40	18,918	77.29	0.34
Smartphone use (per day)	≤1 h	3832	15.33	0.50	5521	22.43	0.43
	1–2 h	3572	14.66	0.37	4997	20.71	0.36
	2–4 h	5078	20.76	0.35	5093	21.20	0.29
	≥4 h	12,299	49.25	0.72	8760	35.66	0.56
Caffeine consumption (per week)	No	19,665	79.34	0.31	18,054	74.08	0.36
	1–2	3475	13.80	0.24	4285	17.41	0.27
	3–4	1032	4.26	0.15	1274	5.32	0.18
	≥5	609	2.60	0.12	758	3.18	0.13

S.E.: Standard Error.

**Table 2 ijerph-16-03285-t002:** Sleep duration and emotion.

Variables		Female			Male		
		*N*	Weighted (%)	S.E. of %	*N*	Weighted (%)	S.E. of %
Sleep duration	<5 h	4863	21.03	0.61	2689	12.26	0.49
	5–6.5 h	9630	40.51	0.50	8044	35.21	0.66
	6.5–8 h	7351	27.68	0.55	8096	31.83	0.56
	≥ 8 h	2937	10.77	0.35	5542	20.70	0.58
Depression	No	17,441	70.28	0.38	19,615	80.16	0.30
	Yes	7340	29.72	0.38	4756	19.84	0.30
Suicidal ideation	No	21,108	85.20	0.29	22,193	90.97	0.22
	Yes	3673	14.80	0.29	2178	9.03	0.22

**Table 3 ijerph-16-03285-t003:** Risk factors for depression.

Variables		Female				Male			
		Weighted (%)	S.E. of %	F value	*p*	Weighted (%)	S.E. of %	F value	*p*
Grade	Middle school 1	24.3712	0.8021	10.4850	<0.001	15.0523	0.6226	16.7712	<0.001
	Middle school 2	29.6874	0.7704			18.3209	0.6880		
	Middle school 3	31.6490	0.8211			19.9142	0.6868		
	High school 1	28.9565	0.8443			19.7825	0.7441		
	High school 2	32.0214	0.9964			22.7324	0.7466		
	High school 3	31.1675	0.9875			22.6568	0.7430		
Economic status	Middle High	28.1690	0.4174	154.9835	<0.001	18.7787	0.3016	105.3032	<0.001
	Low	40.1975	0.9196			27.4379	0.9230		
Living with parents	No	32.2159	3.3586	0.6360	0.07	21.3142	2.9103	0.3129	0.07
	Yes	29.5885	0.3839			19.7398	0.3006		
Current smoking status	No	29.1577	0.3838	126.8878	<0.001	18.5720	0.3028	197.3030	<0.001
	Yes	53.9672	2.3859			34.8478	1.3373		
Current alcohol consumption status	No	27.8907	0.3841	239.0488	<0.001	17.6878	0.3104	307.6788	<0.001
	Yes	42.9252	1.0440			31.0137	0.8161		
Perceived health status	Poor	40.8916	0.6197	679.0039	<0.001	28.1689	0.6186	316.5412	<0.001
	Good	24.3041	0.4081			17.3929	0.3173		
Smartphone use (per day)	≤1 h	25.7723	0.8315	42.0761	<0.001	17.9431	0.5463	19.1407	<0.001
	1–2 h	25.4741	0.7953			18.2405	0.6494		
	2–4 h	27.5461	0.6842			18.7145	0.5952		
	≥4 h	33.1273	0.4777			22.6306	0.4815		
Caffeine consumption (per week)	No	27.6962	0.4138	71.4581	<0.001	17.9650	0.3290	60.4660	<0.001
	1–2	33.9601	0.8588			22.4821	0.7524		
	3–4	43.2046	1.7242			28.0745	1.3770		
	≥5	46.8868	2.1207			35.2576	1.9240		
Sleep duration	<5 h	38.3112	0.8264	80.9660	<0.001	30.0124	0.9729	98.2226	<0.001
	5–6.5 h	30.4133	0.5871			21.7992	0.5702		
	6.5–8 h	25.1332	0.6632			17.7228	0.4485		
	≥8 h	22.1143	0.8286			10.96	0.69		

**Table 4 ijerph-16-03285-t004:** Risk factors for suicidal ideation.

Variables		Female				Male			
		Weighted (%)	S.E. of %	F value	*p*	Weighted (%)	S.E. of %	F value	*p*
Grade	Middle school 1	13.4191	0.6215	13.4146	<0.001	6.8053	0.4394	5.7115	<0.001
	Middle school 2	18.1727	0.7453			10.0402	0.5279		
	Middle school 3	17.3900	0.6331			9.1729	0.4845		
	High school 1	12.4759	0.6305			8.3512	0.5211		
	High school 2	14.3869	0.6563			9.9845	0.5813		
	High school 3	13.0220	0.6540			9.5721	0.5260		
Economic status	Middle High	13.5907	0.2989	156.3675	<0.001	8.2431	0.2185	130.2131	<0.001
	Low	22.9732	0.8383			14.6310	0.6820		
Living with parents	No	14.6383	2.2594	0.0025	0.91	7.1125	1.8856	0.8241	0.91
	Yes	14.7516	0.2930			9.0140	0.2221		
Current smoking status	No	14.4244	0.2880	75.8688	<0.001	8.3902	0.2168	108.8939	<0.001
	Yes	31.0183	2.4854			16.5505	0.9972		
Current alcohol consumption status	No	13.7438	0.2975	112.1103	<0.001	8.0642	0.2249	101.1592	<0.001
	Yes	22.4276	0.9167			14.0194	0.6668		
Perceived health status	Poor	23.6960	0.5474	692.6804	<0.001	16.6240	0.5560	535.4707	<0.001
	Good	10.4881	0.2822			6.7938	0.1960		
Smartphone use (per day)	≤1 h	12.8099	0.6110	24.8611	<0.001	8.7290	0.4517	10.8693	<0.001
	1–2 h	13.0099	0.5998			7.7489	0.4273		
	2–4 h	12.4310	0.5208			8.0185	0.4394		
	≥4 h	16.9506	0.4208			10.5531	0.3655		
Caffeine consumption (per week)	No	13.8106	0.3105	27.4892	<0.001	7.9503	0.2259	54.8799	<0.001
	1–2	16.9768	0.6990			10.3498	0.5178		
	3–4	20.3248	1.3718			12.5059	0.9543		
	≥5	24.4010	1.9508			21.0033	1.6793		
Sleep duration	<5 h	19.5562	0.7021	36.0173	<0.001	14.7645	0.7821	48.7548	<0.001
	5–6.5 h	14.6993	0.4419			9.5070	0.3718		
	6.5–8 h	12.7136	0.4570			8.0591	0.3269		
	≥8 h	11.2509	0.6636			6.2962	0.3610		

**Table 5 ijerph-16-03285-t005:** Factor loading of the pathway analysis for both genders.

Pathway	Female					Male				
	Standardized Factor Loading	Unstandardized Factor Loading	Standard Error	t-Statistics	*p*	Standardized Factor Loading	Unstandardized Factor Loading	Standard Error	t-Statistics	*p*
Sleep duration → Depression	−0.147	−0.111	0.007	–15.15	<0.001	−0.148	−0.105	0.008	–13.23	<0.001
Grade → Depression	−0.069	−0.042	0.007	–5.89	<0.001	−0.055	−0.034	0.007	–4.54	<0.001
Economic status → Depression	0.075	0.238	0.028	8.63	<0.001	0.070	0.223	0.030	7.32	<0.001
Current smoking status → Depression	0.044	0.313	0.062	5.06	<0.001	0.066	0.258	0.045	5.79	<0.001
Current alcohol consumption → Depression	0.087	0.279	0.028	9.90	<0.001	0.103	0.291	0.029	9.93	<0.001
Perceived health status → Depression	−0.181	−0.406	0.019	–21.75	<0.001	−0.127	−0.317	0.021	–14.83	<0.001
Living with parents → Depression	0.011	0.002	0.001	1.53	0.126	0.006	0.001	0.001	0.65	0.518
Caffeine consumption → Depression	0.081	0.049	0.005	9.30	<0.001	0.073	0.040	0.005	8.36	<0.001
Smartphone use → Depression	0.075	0.025	0.003	8.17	<0.001	0.033	0.012	0.004	3.27	0.001
Sleep duration → Suicidal ideation	−0.039	−0.010	0.002	–5.49	<0.001	−0.021	−0.004	0.001	–2.94	0.003
Grade → Suicidal ideation	−0.086	−0.018	0.002	–9.42	<0.001	−0.044	−0.007	0.001	–5.54	<0.001
Economic status → Suicidal ideation	0.040	0.042	0.006	7.00	<0.001	0.030	0.026	0.005	5.26	<0.001
Current smoking → Suicidal ideation	0.021	0.050	0.014	3.69	<0.001	0.020	0.022	0.007	3.31	0.001
Current alcohol consumption → Suicidal ideation	0.024	0.026	0.007	3.75	<0.001	0.010	0.008	0.006	1.44	0.151
Perceived health status → Suicidal ideation	−0.088	−0.067	0.006	–11.54	<0.001	−0.083	−0.057	0.004	–13.39	0.000
Living with parents → Suicidal ideation	0.002	0.000	0.000	0.48	0.635	−0.009	0.000	0.000	–1.43	0.154
Depression → Suicidal ideation	0.400	0.135	0.007	18.52	<0.001	0.391	0.107	0.005	22.11	<0.001
Caffeine consumption → Suicidal ideation	0.016	0.003	0.001	3.17	0.002	0.038	0.006	0.001	7.50	<0.001
Smartphone use → Suicidal ideation	0.007	0.001	0.001	1.13	0.258	−0.002	0.000	0.001	−0.28	0.777

**Table 6 ijerph-16-03285-t006:** Decomposition of direct and indirect effects of sleep duration on suicidal ideation.

	Pathway	Female					Male					F/M
		SF Loading	USF Loading	SE	t-Statistic	*p*	SF Loading	USF Loading	SE	t-Statistic	*p*	Times
**Total**	Sleep duration → Suicidal ideation	−0.098	−0.025	0.002	−10.729	<0.001	−0.079	−0.016	0.002	−9.685	<0.001	1.56
Direct	Sleep duration → Suicidal ideation	−0.039	−0.010	0.002	−5.490	<0.001	−0.021	−0.004	0.001	−2.942	0.003	2.50
Indirect	Sleep duration → Depression → Suicidal ideation	−0.059	−0.015	0.001	−11.956	<0.001	−0.058	−0.011	0.001	−11.245	<0.001	1.36

SF: standardized factor; USF: unstandardized factor; SE: standard error; F/M: more female than male.

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
