# Peer review of "Gender Difference in the Effect of Short Sleep Time on Suicide among Korean Adolescents"

_ijerph, 2019, doi:10.3390/ijerph16183285_

Round 1

Reviewer 1 Report

In this manuscript, the authors explore whether the relationship between sleep disturbance and suicide varies by gender. They conclude that both direct (sleep --> suicidal ideation) and indirect (sleep --> depression --> suicidal ideation) relationships exist between sleep duration and suicidal ideation for both male and female adolescents, but that the direct relationship is much larger for female adolescents.  These findings are important in elucidating the complex relationship between sleep disturbance and suicide in adolescents, but the manuscript could benefit from some edits to more clearly present their findings:

Some editing for English language clarity is required. There are numerous typographical and grammatical errors throughout, and many sentences are phrased in such a way that the authors’ intent is unclear.  Having a native English speaker or person fluent in English read the manuscript for clarity would improve the authors’ communication of these important results. The abstract does not clearly represent what the authors examined in their study. The abstract does not specify that both direct and indirect pathways from sleep duration to suicidal ideation were examined (e.g., states only that the “study was designed to investigate whether there is a difference in the effects of sleep duration on suicide by gender”). As such, when they go on to state “the effect via depressive mood on short sleep time on suicide was not difference between male and female adolescents,” it sounds as though they are reporting a lack of gender differences at all, rather than differences in the direct versus indirect pathways.  This becomes even more confusing when they then describe a different finding (“the direct effect of short sleep time on increasing suicidal ideation is 1.86 times higher in female than in male adolescents”), which seems to contradict the previous sentence as they never specified the difference between direct and indirect paths AND does not reflect the values reported later in the Results and Discussion sections (2.5 times higher for female than male adolescents for the direct path). The abstract then goes on to discuss interpretations of why differences might exist without ever clearly reporting what those differences were (“Differences in the metabolism of sex hormones and sleep associated neurotransmitters might have affected this result, but further studies are needed to clarify the more obvious mechanism. In addition, this result also should be considered when establishing sleep education for adolescents.”).  This space would be better utilized to clearly explain the analyses conducted and results found. In the Introduction, the authors’ literature review makes it seem as though they expect women to be more affected by the relationship between sleep disturbance and suicidal thoughts. The Introduction would be strengthened if they acknowledged this expectation earlier, as opposed to several paragraphs in. It is a reasonable hypothesis supported by the literature they present; might as well just say evidence suggests that gender-differences may disproportionately affect women. Or, if they prefer to leave that unstated, more literature would need to be added to balance the focus on women and men. What about transgender and nonbinary adolescents, a group known to be at particularly high risk for suicide? I recognize that there may not have been any transgender or nonbinary youth in your sample, but if so, this should be acknowledged as a limitation. In the Method section, the authors state “Data from the 13 juveniles who responded to the juvenile health behavior online survey regarding  sleeping time in 2017 were analyzed.” This is REALLY confusing.  I initially believed it to mean that the sample in this study was only 13 youth, but then the subsequent tables seem to have thousands of youth in the sample.  Is the sample actually larger and 13 was a typo?  (Based on the analysis section, it seems as though the sample was substantially larger – 4863 female and 2689 male adolescents endorsing <5 hours sleep per night and 3673 female and 2178 male adolescents endorsing suicidal ideation.)  Or, if the sample was actually 13, that is a VERY small sample – if the sample IS that small, the most likely reason for not finding gender differences would be that the study was not powered to detect differences.  This needs to be clarified (and if the sample was reduced so dramatically, the reason for that attrition should be explained). Given that the original sample was more than 64K students and the analyses focused on only a subset of this (exact N to be determined), do the authors feel that this sample is representative of ALL youth in Korea? How do weighted analyses assist with producing more generalizable results (and how exactly were the analyses weighted)?  Are there any biases or limitations associated with voluntary self-reports surveys that should be acknowledged? The beginning of the Results section presents a lot of data that has nothing to do with gender, sleep, or suicide. It feels as though we are reading a report of the KCDC survey rather than an empirical article testing research hypotheses. These general results would be better reported in a different manuscript detailing the results of the survey; this manuscript proports to be about gender differences in the relationship between sleep and suicide, and thus the results should be focused on testing hypotheses related to those constructs. Demographic info (i.e., age, gender, SES, % living with one parent, etc.) could be moved to the sample description, but all other extraneous information (e.g., smoking, alcohol consumption, smartphone use, etc.) is not relevant to the paper aims. Or, if the authors feel that these factors are relevant risk factors for sleep disorder or suicidal ideation, a more thorough literature review that justifies their inclusion must be provided in the Introduction. Table 3 and 4 are a bit confusing – are these the percentages endorsing each risk factor (e.g., caffeine, sleep, etc.) for only those youth who endorsed depression (Table 3) and suicidal ideation (Table 4)? Were these all univariate logistic regressions, or were all of these predictors ever entered into a large logistic regression to see which remained significant when tested against each other?  Do the p-values represent significant prediction of the outcome by the factor, significant differences between gender, or both (mixed-model analyses)?  More detail on how analyses were conducted and what the values in the tables represent would be helpful.  It would also be useful to include the test statistic (F etc., not just the p-value), in the table and/or enter the statistical values in-text. As written, it is hard to tell how analyses were ran and if the methodology was appropriate. The way the grades are described (“Grade 1 to 6” when actually referring to youth in middle and high school) was confusing. Although this is mentioned in the Method section, I would consider renaming the grades to something like Middle School 1, 2, 3 and High School 1, 2, 3 to clarify.  As written, I initially though youth in grade 2 of elementary school (e.g., 7-8 years old) had the highest rates of suicidal ideation and was baffled.  The results make a bit more sense when realizing that “Grade 2” is actually the second year of middle school (equivalent to junior high or freshman year of high school in the US).  Given that international audiences may not be as familiar with the Korean school structure, I would consider giving the grades more intuitive names throughout and including the ages of youth associated with each grade somewhere in the Method section. Consider reformatting to landscape orientation or using abbreviations for Tables 5 and 6 so that words do not break across multiple lines in the header. It would be helpful if standard errors for the path estimates were added to figure 1. It is pretty standard to include both the parameter estimate and some indicator of variance. Does Table 5 imply that all risk factors were covaried in the path analysis presented in figure 1? It looks like parameters were estimated for prediction of both depression and suicidal ideation, but in the figure the risk factors are only depicted as being controlled for the relationship between depression and suicidal ideation?  A more clearly-defined analysis section would help with interpretation of the results. Ultimately, the authors conclude that sleep duration’s relationship with suicidal ideation was partially mediated by sleep duration’s effect on depression; the direct relationship between sleep duration and suicidal ideation was greater in female than male adolescents, while no gender difference was present in the indirect relationship (sleep --> depression --> suicidal ideation). Was this controlling for all the other risk factors listed?

Author Response

Dear sir

I appreciate to your affectionate paper review.

This study was based on KCDC data, which could represent Korean adolescents. According to the results of previous studies, the health behavior and emotional state of Korean adolescents do not seem to be significantly different from the behavior of adolescents in other countries.

As described in this paper, sleep time directly affects depression and suicidal thoughts. Our results showed that these effects on suicide were more potent in females than mal adolescents.

Based on the results of this paper, we need to pay more attention to adolescent sleep education. And this study found that education requires a different approach in male and female adolescents.

We have modified or added the manuscript all of your detailed reviews

Thanks a lot, again.

Some editing for English language clarity is required. There are numerous typographical and grammatical errors throughout, and many sentences are phrased in such a way that the authors’ intent is unclear.  Having a native English speaker or person fluent in English read the manuscript for clarity would improve the authors’ communication of these important results. We got English editing service in IJERPH editing. The abstract does not clearly represent what the authors examined in their study. The abstract does not specify that both direct and indirect pathways from sleep duration to suicidal ideation were examined (e.g., states only that the “study was designed to investigate whether there is a difference in the effects of sleep duration on suicide by gender”). As such, when they go on to state “the effect via depressive mood on short sleep time on suicide was not difference between male and female adolescents,” it sounds as though they are reporting a lack of gender differences at all, rather than differences in the direct versus indirect pathways.  This becomes even more confusing when they then describe a different finding (“the direct effect of short sleep time on increasing suicidal ideation is 1.86 times higher in female than in male adolescents”), which seems to contradict the previous sentence as they never specified the difference between direct and indirect paths AND does not reflect the values reported later in the Results and Discussion sections (2.5 times higher for female than male adolescents for the direct path). The abstract then goes on to discuss interpretations of why differences might exist without ever clearly reporting what those differences were (“Differences in the metabolism of sex hormones and sleep associated neurotransmitters might have affected this result, but further studies are needed to clarify the more obvious mechanism. In addition, this result also should be considered when establishing sleep education for adolescents.”).  This space would be better utilized to clearly explain the analyses conducted and results found. In the Introduction, the authors’ literature review makes it seem as though they expect women to be more affected by the relationship between sleep disturbance and suicidal thoughts. I edited and correct abstract following your comments. A close association between the duration of sleep and suicide has been reported in previous studies. This study was designed to investigate whether there is a difference in the effects of sleep duration on suicide by gender. This study was conducted based on the results of a volunteer online survey for adolescents in middle and high school in the Republic of Korea. The results showed that the effect of a depressive mood on short sleep time and on suicide was not different between male and female adolescents. It has been reported that the direct effect of short sleep time on increasing suicidal ideation is 50 times higher in female than in male adolescents. Differences in the metabolism of sex hormones and sleep associated neurotransmitters might have affected this result, but further studies are needed to clarify more obvious mechanisms. In addition, this result should be considered when establishing sleep education for adolescents.

The Introduction would be strengthened if they acknowledged this expectation earlier, as opposed to several paragraphs in. It is a reasonable hypothesis supported by the literature they present; might as well just say evidence suggests that gender-differences may disproportionately affect women. Or, if they prefer to leave that unstated, more literature would need to be added to balance the focus on women and men. What about transgender and nonbinary adolescents, a group known to be at particularly high risk for suicide? I recognize that there may not have been any transgender or nonbinary youth in your sample, but if so, this should be acknowledged as a limitation. We hand revised the introduction to be more concise as you require and expect the results of this study. When conducting this survey in Korea, questionnaire about transgender or nobinary youth were not included. In the Method section, the authors state “Data from the 13 juveniles who responded to the juvenile health behavior online survey regarding  sleeping time in 2017 were analyzed.” This is REALLY confusing.  I initially believed it to mean that the sample in this study was only 13 youth, but then the subsequent tables seem to have thousands of youth in the sample.  Is the sample actually larger and 13 was a typo?  (Based on the analysis section, it seems as though the sample was substantiallylarger – 4863 female and 2689 male adolescents endorsing <5 hours sleep per night and 3673 female and 2178 male adolescents endorsing suicidal ideation.)  Or, if the sample was actually 13, that is a VERY small sample – if the sample IS that small, the most likely reason for not finding gender differences would be that the study was not powered to detect differences.  This needs to be clarified (and if the sample was reduced so dramatically, the reason for that attrition should be explained). This is the 13th study, and as you say, there is confusion, so we corrected it to 13th. Given that the original sample was more than 64K students and the analyses focused on only a subset of this (exactN to be determined), do the authors feel that this sample is representative of ALL youth in Korea? How do weighted analyses assist with producing more generalizable results (and how exactly were the analyses weighted)?  Are there any biases or limitations associated with voluntary self-reports surveys that should be acknowledged? This study was designed to understand the current state of mental health of Korean adolescents by Korean government. For this reason, this data is representative of Korean adolescents. However, the sampling method is based on middle / high school, and the adolescents who do not go to school are excluded. The characteristics of the students could vary according to the characteristics of the school. For example, there might be school characteristics such as gifted schools and specialized high school for specific jobs. The beginning of the Results section presents a lot of data that has nothing to do with gender, sleep, or suicide. It feels as though we are reading a report of the KCDC survey rather than an empirical article testing research hypotheses. These general results would be better reported in a different manuscript detailing the results of the survey; this manuscript proports to be about gender differences in the relationship between sleep and suicide, and thus the results should be focused on testing hypotheses related to those constructs. Demographic info (i.e., age, gender, SES, % living with one parent, etc.) could be moved to the sample description, but all other extraneous information (e.g., smoking, alcohol consumption, smartphone use, etc.) is not relevant to the paper aims. Or, if the authors feel that these factors are relevant risk factors for sleep disorder or suicidal ideation, a more thorough literature review that justifies their inclusion must be provided in the Introduction. We reviewed that smoking, alcohol, etc. could also affect suicidal ideation and depression based on previous studies. We added some previous research in introduction part to explain it. Table 3 and 4 are a bit confusing – are these the percentages endorsing each risk factor (e.g., caffeine, sleep, etc.) for only those youth who endorsed depression (Table 3) and suicidal ideation (Table 4)? Were these all univariate logistic regressions, or were all of these predictors ever entered into a large logistic regression to see which remained significant when tested against each other?  Since this data was extracted under the complex sampling design of the KCDC health surbey, weights, stratification variables, and colony variables were selected according to the analysis guidelines distributed by KCDC. We used a complex sample desing analysis method (PROC SURVEYFREQ, SAS 9.3 version) in Table 3/4 We added F value on Table 3 and 4. Do the p-values represent significant prediction of the outcome by the factor, significant differences between gender, or both (mixed-model analyses)?  More detail on how analyses were conducted and what the values in the tables represent would be helpful.  It would also be useful to include the test statistic (F, not just the p-value), in the table and/or enter the statistical values in-text. I added the value of F and p-value.

As written, it is hard to tell how analyses were ran and if the methodology was appropriate. The way the grades are described (“Grade 1 to 6” when actually referring to youth in middle and high school) was confusing. Although this is mentioned in the Method section, I would consider renaming the grades to something like Middle School 1, 2, 3 and High School 1, 2, 3 to clarify. As written, I initially though youth in grade 2 of elementary school (e.g., 7-8 years old) had the highest rates of suicidal ideation and was baffled.  The results make a bit more sense when realizing that “Grade 2” is actually the second year of middle school (equivalent to junior high or freshman year of high school in the US).  Given that international audiences may not be as familiar with the Korean school structure, I would consider giving the grades more intuitive names throughout and including the ages of youth associated with each grade somewhere in the Method section. à We correct 1-6 grade to middle school 1,2,3 and High school 1,2,3

Consider reformatting to landscape orientation or using abbreviations for Tables 5 and 6 so that words do not break across multiple lines in the header. The table 5,6 has been modified following your recommendations. It would be helpful if standard errors for the path estimates were added to figure 1. It is pretty standard to include both the parameter estimate and some indicator of variance. The figure was corrected by your recommendation. The modified figure seems to be easier to understand. Does Table 5 imply that all risk factors were covaried in the path analysis presented in figure 1? It looks like parameters were estimated for prediction of both depression and suicidal ideation, but in the figure the risk factors are only depicted as being controlled for the relationship between depression and suicidal ideation?  A more clearly-defined analysis section would help with interpretation of the results. Ultimately, the authors conclude that sleep duration’s relationship with suicidal ideation was partially mediated by sleep duration’s effect on depression; the direct relationship between sleep duration and suicidal ideation was greater in female than male adolescents, while no gender difference was present in the indirect relationship (sleep --> depression --> suicidal ideation). Was this controlling for all the other risk factors listed? We reported that the effects of short sleep time on suicidal ideation (compared to the most well known depression) when controlling for other variables are differences between men and women. The conclusion of this study is that direct impacts on suicide of short sleep duration were more potent in women.

Reviewer 2 Report

This study reports on a volunteer online survey of a large sample of middle and high school Korean adolescents (n=49, 152). The results found an interesting association between short sleep times increasing suicide ideation in female versus male adolescents. However, the report needs to address the following issues:

The title is misleading should be revised to “Gender difference in the effect of short sleep time on suicide ideation among Korean adolescents”.  Line 45 on page 2 indicated that the study was conducted “from the 13 juveniles” but I believe this number is incorrect given the reported sample size. On page 2, lines 75, 76, I am unclear how sleep duration and smartphone use were analyzed as continuous variable when they are described as being categorical variables? In the discussion lines 165, 166, the suggestion that males adapt to stress situation caused by sleep deprivation better than females seems unlikely. More plausible explanations are that males compared to females develop other psychopathologic outcomes other than suicide ideation such as more substance abuse, aggression or death by suicide than females. In the discussion lines 175 and 179 appear to contradict each other; are females more vulnerable to increased or decreased leptin levels? A major limitation of the survey was the use of cross-sectional data. Therefore, the nature of the relationship between sleep duration, suicide ideation and depression is unknown. Of course, the causal relationship may be reversed in that depressive symptoms or suicide ideation lead to reduced sleep time. Referring to the possible preventive implications of these findings rather than speculating on possible biological mechanism would strengthen the discussion. Would shortened sleep time be a useful risk indicator for female adolescents with suicide ideation? The manuscript needs a very careful editing to correct several grammatical errors. The authors should reduce the number and size of the tables.

Author Response

Dear sir

I appreciate to your affectionate paper review.

This study was based on KCDC data, which could represent Korean adolescents. According to the results of previous studies, the health behavior and emotional state of Korean adolescents do not seem to be significantly different from the behavior of adolescents in other countries.

As described in this paper, sleep time directly affects depression and suicidal thoughts. Our results showed that these effects on suicide were more potent in females than mal adolescents.

Based on the results of this paper, we need to pay more attention to adolescent sleep education. And this study found that education requires a different approach in male and female adolescents.

We have modified or added the manuscript all of your detailed reviews

Thanks a lot, again.

The title is misleading should be revised to “Gender difference in the effect of short sleep time on suicide ideation among Korean adolescents”.  Line 45 on page 2 indicated that the study was conducted “from the 13 juveniles” but I believe this number is incorrect given the reported sample size.

This is the 13th survey that conducted in Korea, not 13 adolescents. So revised to 13th.

On page 2, lines 75, 76, I am unclear how sleep duration and smartphone use were analyzed as continuous variable when they are described as being categorical variables?

Sleep time and smartphone use time are categorical variables, not continuous variables. We corrected it, following your recommendations.

 In the discussion lines 165, 166, the suggestion that males adapt to stress situation caused by sleep deprivation better than females seems unlikely. More plausible explanations are that males compared to females develop other psychopathologic outcomes other than suicide ideation such as more substance abuse, aggression or death by suicide than females.

I agree with your opinion. This sentence was ambiguous. So, I changed this sentence blow for clearity.

The development of a defensive mechanism related hormones in males, may have helped males to adapt to the various emotional stress situation including sleep deprivation.17

 In the discussion lines 175 and 179 appear to contradict each other; are females more vulnerable to increased or decreased leptin levels?

This sentence was ambiguous. So, I changed this sentence blow for clearity.

Female gender is more increased levels of morning plasma leptin after sleep restriction than males.21

 A major limitation of the survey was the use of cross-sectional data. Therefore, the nature of the relationship between sleep duration, suicide ideation and depression is unknown. Of course, the causal relationship may be reversed in that depressive symptoms or suicide ideation lead to reduced sleep time. Referring to the possible preventive implications of these findings rather than speculating on possible biological mechanism would strengthen the discussion.

I agree with your opinion and recommendation So, I added this sentence in Discussion part.

Third, this is cross sectional questionnaire based design, there could be fatal limitation to clarify the relationship between cause and outcome. We used the path analysis method to compensate this limitation. Follow up studies suggest that this limitation could be overcome.

Would shortened sleep time be a useful risk indicator for female adolescents with suicide ideation?

In several similar studies, shorter sleep time, both gender, have been reported as risk factors for suicide and depression. In this study, I want to show that the effects of short sleep time are stronger in girls than boys.

The manuscript needs a very careful editing to correct several grammatical errors. The authors should reduce the number and size of the tables. 

We already have a native speaker review this paper. However, due to readability and grammar errors, we applied for and reviewed the MDPI English editing service.

Round 2

Reviewer 2 Report

The authors have addressed the reviewer's concerns.